


# Using Respiration Quotients to Track Changing Sources of Soil Respiration Seasonally and with Experimental Warming

Caitlin Hicks Pries[1,2], Alon Angert[3], Cristina Castanha[2], Boaz Hilman[3,4], and Margaret S. Torn[2]

[1]Department of Biological Sciences, Dartmouth College, Hanover, NH, 03784, United States of America
[2] Climate and Ecosystem Science Division, Earth and Environmental Science Area, Lawrence Berkeley National Laboratory, Berkeley, CA, 94720, United States of America
[3]The Institute of Earth Sciences, The Hebrew University of Jerusalem, Givat-Ram, Jerusalem 91904, Israel
[4]Currently at: Department of Biogeochemical Processes, Max-Planck Institute for Biogeochemistry, Jena, 07745, Germany

*Correspondence to*: Caitlin Hicks Pries (Caitlin.Pries@dartmouth.edu)

**Abstract.** Developing a more mechanistic understanding of soil respiration is hampered by the difficulty in determining the contribution of different organic substrates to respiration and in disentangling autotrophic versus heterotrophic and aerobic versus anaerobic processes. Here, we present a relatively novel tool for better understanding soil respiration: the apparent respiration quotient (ARQ). ARQ is the amount of $CO_2$ produced in the soil divided by the amount of $O_2$ consumed and it changes according to which organic substrates are being consumed and whether oxygen is being used as an electron acceptor. We investigated how the ARQ of soil gas varied seasonally, by soil depth, and by experimental warming *in situ* in a coniferous forest whole-soil-profile warming experiment over two years. We then compared the patterns in ARQ to those of soil $\delta^{13}CO_2$. Our measurements showed strong seasonal variations in ARQ from ≈0.9 during the late spring and summer to ≈0.7 during the winter. This pattern likely reflected a shift from respiration being fueled by oxidized substrates like sugars and organic acids derived from root and root respiration during the growing season to more reduced substrates such as lipids and proteins derived from microbial necromass during the winter. This interpretation was supported by $\delta^{13}CO_2$ values, which were relatively depleted, like lipids, in the winter and more enriched, like sugars, in the summer. Furthermore, wintertime ARQ was higher in warmed (+4°C) than in control plots, probably due to an increase in the use of more oxidized carbon substrates with warming. Our results demonstrate that soil ARQ shows strong seasonal patterns in line the phenology of carbon inputs and patterns in soil $\delta^{13}CO_2$, verifying ARQ as a tool for disentangling the biological sources contributing to soil respiration.





**1 Introduction**

Despite making extensive measurements of soil respiration (Bond-Lamberty and Thomson, 2010), scientists lack methods to disentangle the processes underlying, and substrates contributing to, soil respiration, which hampers predictions of terrestrial carbon cycle responses to global change (Phillips et al., 2017). Mechanistic uncertainty surrounding soil respiration is partly responsible for the 1000 Pg spread in model predictions of end-of-century terrestrial carbon-climate feedbacks (Friedlingstein et al., 2013). Soil respiration is the sum of autotrophic respiration by plant roots and heterotrophic respiration by soil microbes. Heterotrophic respiration, which has increased globally over the past three decades (Bond-Lamberty et al., 2018), is itself the sum of various processes using different sources of energy. For example, microbes consume different organic substrates depending on what molecules are accessible and whether the microbes are living in the rhizosphere or bulk soil, and microbes utilize different terminal electron acceptors depending on $O_2$ availability in the microsites in which they reside (Keiluweit et al., 2016; Liptzin et al., 2011). The electron donors (the organic substrates) and the electron acceptors used by soil microbes during respiration cannot be resolved by measuring the $CO_2$ flux alone. Previous studies have used measurements of $\delta^{13}C$ to partition respiration into autotrophic and heterotrophic components (e.g., Dorrepaal et al., 2009), radiocarbon to partition respiration sources by age (e.g., Trumbore, 2000), or both isotopes in combination to more finely separate respiration among sources (e.g., Hicks Pries et al., 2013; Hopkins et al., 2012). However, isotopes are not the only way to disentangle soil respiration's various components.

Our ability to understand soil respiration is limited by measuring only one half of the respiration equation, the $CO_2$ produced. Simultaneously measuring the $O_2$ consumed can provide a more mechanistic understanding of the processes and substrates contributing to soil respiration (Phillips et al., 2017). The paired measurements of $CO_2$ and $O_2$ can be used to calculate a respiration quotient (ARQ; Angert and Sherer, 2011). All organic matter has an oxidative ratio (OR=1/RQ), which can be calculated based on an elemental analysis of its C, H, O, and N (Masiello et al., 2008). The oxidation state of carbon in carbohydrates is 0 with a corresponding RQ of 1. More reduced energy sources such as lipids have lower RQ values (≈0.73) and the RQ of proteins range from 0.67 to 1; more oxidized sources such as organic acids have RQ ranges from 1 to 4 (Masiello et al., 2008; Table 1). The RQ of aerobic respiration therefore changes based on what substrates are being consumed (Dilly, 2001; Theenhaus et al., 1997). Anaerobic respiration increases RQ to values greater than one, as electron acceptors like Fe(III) and $NO_3^-$ replace $O_2$. Thus, RQ can help differentiate between the electron donors (organic substrates) and terminal electron acceptors used during soil respiration. We will refer to 'apparent' RQ because not all ecosystem $CO_2$ or $O_2$ fluxes are due to respiratory processes (Angert and Sherer, 2011). For example, fluctuating redox conditions can lead to consumption of $O_2$ during metal oxidation and drive ARQ below the value of the most reduced organic matter (Angert et al., 2015).



Thus far, CO2:O2 ratios have been primarily used to understand large scale earth system processes and only few studies have
examined processes within ecosystems. This ratio in atmospheric samples has been used to estimate a) the magnitude of the
terrestrial carbon sink, because carbon uptake by terrestrial ecosystems is balanced by $O_2$ production whereas ocean $CO_2$
uptake is decoupled from $O_2$, (Keeling, 1988; Keeling et al., 1996; Randerson et al., 2006; Worrall et al., 2013) and b)
anthropogenic impacts on the carbon cycle, based on the principle that burning of reduced fossil fuels results in a different
oxidative ratio than does photosynthesis and subsequent respiration of carbohydrates (Keeling, 1988). The CO2:O2 ratio of
ecosystem-atmosphere exchanges is an essential quantity in these carbon cycle calculations. CO2:O2 ratios have been estimated
from measurements of net ecosystem exchange of $CO_2$ and $O_2$ (e.g., Seibt et al., 2004) and from elemental analysis of biomass
(Hockaday William C. et al., 2015; e.g., Masiello et al., 2008), both of which are assumed to be similar over multiyear
timescales. In early carbon sink calculations, the oxidative ratio of ecosystem fluxes was assumed to be 1.1 (ARQ=0.9) based
on a single study of temperate soils (Severinghaus, 1995). However, the few subsequent studies examining the CO2:O2 ratio
of soil respiration fluxes have shown soil fluxes can deviate widely from that value.

Soil ARQ from incubations shift as a result of temperature changes, substrate additions, and soil management. For example,
the ARQ of peat soils decreased from about 1.1 to about 0.6 when temperatures increased from 0 to 20°C, attributed to changing
substrate use (Chapman and Thurlow, 1998). Glucose additions to German forest soils increased soil ARQ to 0.95-1.0 from a
basal value around 0.7 (Dilly, 2001; Theenhaus et al., 1997). Soils under organic agriculture were found to have a greater ARQ
(1.19) than soils under conventional agriculture (0.72; Theenhaus et al., 1997). Soil ARQ in mesocosms containing pine
seedlings changed seasonally and when the pine seedlings were cut, indicating the ratio is responsive to changes in vegetation
(Andersen and Scagel, 1997; Scagel and Andersen, 1997). Lastly, in one of the only studies using *in situ* measurements, soil
ARQ taken from gas wells across multiple forested ecosystems ranged widely from 0.14 to 1.23 indicating the influence of
abiotic processes that consume $O_2$ (Angert et al. 2015). The wide range in soil ARQ values associated with different
biochemical conditions indicates the ratio has the potential to provide insight into the substrates contributing to respiration as
well as into abiotic $O_2$ consumption. Finer scale research is needed, however, to explore ARQ values in the sam soils under
different conditions to learn what these values indicate about the processes and substrates contributing to soil respired $CO_2$.
Here we investigated how the ARQ of soil gas *in situ* varied seasonally, by soil depth, and by experimental warming in a
whole-soil-profile warming experiment in a well-drained, oxygenated coniferous forest soil (Hicks Pries et al., 2017). We
characterized soil ARQ at 30 and 90 cm depths in the winter and growing season over two years and compared the patterns in
ARQ to monthly patterns in soil profile $\delta^{13}CO_2$. We hypothesized that ARQ values would change seasonally and with warming
reflecting the values of the organic carbon substrates being consumed by microbes. Like ARQ, the $\delta^{13}C$ of soil $CO_2$ is
influenced by the use of different organic substrates since more reduced substrates tend to also be depleted in $\delta^{13}C$ (Bowling
et al., 2008). By comparing ARQ values to other indicators of respiration sources, such as $\delta^{13}C$, augmented by what we
understand about plant allocation of carbon substrates belowground, we aim to advance the utility of ARQ as a tracer of
respiration processes.



## 2 Methods

### 2.1 Warming Experiment

The whole soil profile warming experiment is located at the University of California Blodgett Forest Research Station, in the Sierra Nevada foothills near Georgetown, CA at 1370 m above sea level. Mean annual precipitation is 1774 mm with most of it occurring from November through April and mean annual temperature is about 12.5°C (Bird and Torn, 2006). The experiment is in a thinned 80-year-old stand of mixed conifers including ponderosa pine (*Pinus ponderosa*), sugar pine (*Pinus lambertiana*), incense cedar (*Calodefrus decurrens*), white fir (*Abies concolor*), and douglas fir (*Pseudotsuga menziesii*). The soils are Holland series: fine-loamy, mixed, superactive, mesic Ultic Haploxeralfs of granitic origin with thick, >5 cm O horizons, minimal carbonates (Rasmussen et al., 2005), and a pH that ranges from 5.6 to 6.5 (Hicks Pries et al., 2018). The warming treatment warmed the soil +4°C to 1 m depth while maintaining the natural temperature gradient with depth and temporal variations in soil temperature as described in Hicks Pries et al. (2017). Briefly, there were three pairs of control and heated 3 m diameter circular plots heated by 22 vertical resistance heater cables in metal conduit (BriskHeat, Ohio, USA) that surrounded them. To compensate for surface heat loss, two concentric rings of heater cable at 1 and 2 m in diameter were installed 5 cm below the soil surface in heated plots. Unheated cables were installed similarly in control plots. Heating throughout the plot volume was generally even, ranging from 3.5 to 4.5°C except at 5 cm depth where the heated plots were on average only $2.4 \pm 1.2$°C warmer than the control due to a lack of aboveground heating. Soil moisture was slightly decreased in the warmed plots by an average of 1.5-3.5% volumetric water content (Hicks Pries et al., 2017).

### 2.2 Sample Collection and Analysis

Dataloggers (CR1000, Campbell Scientific, Utah, USA) continuously recorded soil temperature and moisture at 30 min intervals. Temperature was monitored at 5, 15, 30, 50, 75, and 100 cm depths at a radial distance of 0.75 m from the center of each plot. Temperature probes consisted of thermistors (Omega 44005) epoxied to PVC rods, placed inside thin-walled steel conduit. To monitor soil moisture, we used an enviroSCAN (Sentek, Australia) probe fitted with capacitance sensors at 10, 30, 50, and 90 cm at a radial distance of 0.75 m from the center of each plot. We calibrated the soil moisture measurements by comparing the sensor values at each depth to the volumetric water content measured in nearby (within 0.5 m) soil cores that were sampled five times over two years.

Each of the six plots has a set of gas wells at 15, 30, 50, 75, and 90 cm. The gas wells were 6.35 mm diameter stainless steel tubes inserted into the soil at a 45° angle to the desired depth and topped with straight swage pipefittings (Swagelok Ohio, USA) with septa. For $CO_2$ and $\delta^{13}CO_2$ measurements, samples were collected from the wells with a syringe on a nearly monthly basis from March 2014 through June 2017 (32 months total) and always during morning hours. After clearing the headspace in each well, a 25 ml gas sample was transferred to an evacuated 20 ml septum-topped glass vial. For analysis, 5 ml samples were injected into the small sample isotope module of a cavity ring down spectrometer (CRDS, Picarro, Santa Clara,





California) where they were diluted with ultra zero air (without $CO_2$). A four-point calibration curve ranging from 2,000 to
20,000 ppm ($\delta^{13}C$=-26.7‰) was used to calculate the $CO_2$ concentration from the CRDS data and to correct for mass
dependency of the $\delta^{13}C$ measurement.

In July 2015, February 2016, April 2016, August 2016, March 2017, and June 2017, we collected additional samples from the
30 (except July 2015) and 90 cm gas wells into 13 ml flasks equipped with O-ring valves (LouwersHanique, Hapert,
Netherlands) to simultaneously measure $CO_2$ and $O_2$ concentrations in order to calculate ARQ. The flasks were analyzed in
the laboratory at the Hebrew University by a closed system (The Hampadah; Hilman and Angert, 2016). This fully automated
system uses an infra-red gas analyzer (IRGA) for $CO_2$ measurement (LI 840A LI-COR; Lincoln, NE, USA) and a fuel-cell
based analyzer (FC-10; Sable Systems International, Las Vegas, NV, USA) for measuring $O_2$. The flasks were analyzed within
2-3 weeks of collection.

In June 2017, we also ran a set of short (3 hour) incubations of root-free soil and of excised roots collected adjacent to the
experimental plots. We collected four mineral soil cores with a 5 cm diameter hammer corer, separated the cores into 0-20 and
20-40 cm depths, and removed roots >1 mm diameter. Roots were collected from four 25 cm x 25 cm x 25 cm soil pits. We
rinsed roots with water to remove soil and blotted them dry before placing them into mason jars. The root-free soil was also
placed into mason jars, and both sets of mason jars were flushed with ambient, outside air. After a three-hour incubation of the
root samples and a 21-hour incubation of the soil samples, the headspace was sampled for $CO_2$ and $O_2$ and analyzed as
described above. Incubations were run at room temperature, which was similar to the field temperature at the time of collection.
**2.3 Sample Calculations and Statistics**
To calculate ARQ, we used the following equation from Angert et al. (2015):
$$ARQ = -0.76\frac{\Delta CO_2}{\Delta O_2}$$

Where ARQ is the apparent respiratory quotient, $\Delta CO_2$ (ppmv) is the difference between $CO_2$ concentrations in the soil pore
space gas and ambient (i.e., near-surface aboveground) samples, $\Delta O_2$ (ppmv) is the difference of the soil pore space $O_2$
concentration and ambient $O_2$ concentration, and 0.76 is the ratio of $CO_2$ to $O_2$ diffusivity in air (Massman, 1998). The negative
sign is for convenience so the ARQ value will typically be positive, because the difference in $O_2$ concentration is always
negative. For the jar incubations we used the same equation without the 0.76 factor. Ambient $CO_2$ concentrations were
measured in the field at the time of sampling with the CRDS, while the ambient $O_2$ concentration was assumed to be 20.95%.
To relate the $\delta^{13}C$ value of soil pore space $CO_2$ to the $\delta^{13}C$ of $CO_2$ production, we corrected the pore-space $\delta^{13}C$ value for
diffusion since $^{13}C$ diffuses slower in air than $^{12}C$ and thus the measured value does not accurately represent the value of
production. For the correction, we used the following equation from Bowling et al. (2015):



$$\delta_{production} = \frac{C_s(\delta_s - 4.4) - C_a(\delta_a - 4.4)}{1.0044(C_s - C_a)}$$

Where $C_s$ is the soil pore space $CO_2$ concentration (ppmv), $\delta_s$ (‰) is the isotopic composition of soil pore space $CO_2$ and $C_a$
and $\delta_a$ are the $CO_2$ concentration and isotopic composition of ambient air, respectively. The ambient $CO_2$ concentrations and
$\delta^{13}C$ values needed for these corrections were measured in the field at the time of sampling with the CRDS.

To investigate the effects of season, warming treatment, and soil depth on ARQ and $\delta^{13}C$, we ran multiple regressions in R (R
Development Core Team, 2017). Because ARQ was not sampled from both depths on all dates, we ran separate regressions
for each depth (30 and 90 cm) and then ran a regression that included a depth effect while dropping the first sampling date. In
all regressions, treatment and sampling date (as a factor) were fixed effects. Following Zuur et al. (2009), we used a full model
with all fixed effects and their interactions to optimize the random effects and autocorrelation structure based on AIC. For both
versions, we used the individual gas well as a random effect and a temporal autocorrelation did not improve the model, nor
did an autocorrelation function graph indicate one was needed. We chose the significant fixed effects by performing a series
of pairwise model comparisons using AIC and the $F$ test, dropping the least significant variables each time until only variables
that improved the model fit remained. The p-values reported are those from the t-tests of the summary.lme function of best fit
model. To investigate seasonal patterns in $\delta^{13}CO_2$, we had more data in terms of both length of time and temporal density of
sampling and were thus able to treat month as a continuous variable. We fit a sine function and tested models including the
first and second harmonics of the month effect as well as linear fixed effects of depth, treatment, and a depth by treatment
interaction. Graphical exploration indicated the sinusoidal pattern differed slightly by year, so we also added a year effect to
the second harmonic of the month effect. As above, we used the full fixed effect model to test the best random and
autocorrelation structure. Individual gas well depth was used as a random effect and an autocorrelation-moving average
(corARMA, p=2, q=2) was the best correlation structure. To test relationships between ARQ and $\delta^{13}CO_2$, and both ARQ and
$\delta^{13}CO_2$ individually versus soil temperature and volumetric water content, we ran mixed-model regressions with individual
gas well as a random effect. We used sensor data from the depths sampled by gas wells, which limited our analyses to the 30
and 90 cm sensor depths only. We tested the need for autocorrelation structures based off of AIC and none improved the
model. For these regressions, we report pseudo $R^2$ values calculated from a linear regression of the actual data versus the model
predicted data. For all models, we graphically checked the residuals for violations of normality and heterogeneity of variance.
For $\delta^{13}CO_2$ analyses, we dropped the 15 cm depths due to their unusually low $\delta^{13}C$ value (<-32‰) after correction (Eq. 1),
which indicated potential intrusion of atmospheric air during sampling that led to an overcorrection. We used one-way
ANOVA's to compare the ARQ of soil and root incubations and the ARQ of two soil depths we incubated. All statistics were
performed in R v 3.4.1 and regressions were done using the lme function (R Development Core Team, 2017).





## 3 Results

Both ARQ and $\delta^{13}CO_2$ had similar, strong seasonal patterns (Fig. 1a and 1b). ARQ values were higher during the growing season ($0.89 \pm 0.01$) and lower during the winter ($0.70 \pm 0.02$). In ARQ regression analyses for both depths, there was a significant effect of date ($p<0.0001$) with February 2016 and March 2017 differing significantly from July 2015 (90 cm only), April 2016, August 2016, and June 2017. Similarly, $\delta^{13}C$ was more enriched during the summer (June through October, $-28.43 \pm 0.08$) and more depleted during the winter and spring (November through May, $-29.26 \pm 0.05$). While individual dates were not compared statistically for $\delta^{13}CO_2$, the vast improvement in model fit using month as a sine function instead of a linear function ($\Delta AIC=112$) is strong statistical evidence for a seasonal effect (Fig. 2b). ARQ and $\delta^{13}CO_2$ were significantly related according to the mixed effect regression model (Fig. 1c, $p<0.0001$, $n=64$, pseudo $R^2=0.20$). However, the patterns in ARQ and $\delta^{13}CO_2$ did not match during April.

Both ARQ and $\delta^{13}CO_2$ differed by warming treatment (Fig. 2) and by depth (Table 1), although to a lesser magnitude than by seasonal differences. For 30 cm depths, there was a significant treatment-by-date interaction ($p=0.051$, $n=30$) whereby heated plots had greater ARQ values during the winter months (February 2016 and March 2017; Fig. 2a). In contrast, the best fit model for 90 cm did not include a significant treatment effect or treatment-by-date interaction (Fig. 2a). For $\delta^{13}CO_2$, treatment was a significant effect ($p=0.0065$, $n=758$) with warmed soil on average having a slightly more enriched $\delta^{13}CO_2$ ($-28.33 \pm 0.05$) than the control soil ($-28.83 \pm 0.06$; Fig 2b). The treatment-by-depth interaction was not significant for $\delta^{13}CO_2$ and was not included in the best fit model. Looking at depth only, ARQ at 30 cm was significantly greater than ARQ at 90 cm by 0.03 units ($p=0.0495$, $n=59$), while $\delta^{13}CO_2$ became slightly more enriched with depth going from $-28.98$ at 30 cm to $-28.34$ at 90 cm ($p=0.0089$, $n=758$).

Both ARQ and $\delta^{13}CO_2$ showed strong relationships with soil climate (Fig 3). We tested relationships with soil temperature and soil moisture individually because of the strong negative correlation between temperature and moisture in this Mediterranean climate (pearson's $r=-0.76$ to $-0.78$). ARQ increased significantly with increasing soil temperatures ($p<0.0001$, $n=64$, pseudo $R^2=0.46$; Fig 3a). It decreased with increased soil moisture, but the fit ($p<0.0001$, $n=59$ due to missing VWC values, pseudo $R^2=0.18$) was not as good as for temperature (Fig. 3b). $\delta^{13}CO_2$ became more enriched with increasing soil temperatures ($p<0.0001$, $n=375$, pseudo $R^2=0.33$; Fig. 3c) and more depleted with increased soil moisture ($p<0.0001$, $n=345$ due to missing VWC values, pseudo $R^2=0.30$; Fig. 3d).

Our incubations of roots and of root-free soil indicated that heterotrophic and autotrophic respiration has significantly different ARQ values, at least during the summer when we performed the incubations. Roots had a greater ARQ ($0.87 \pm 0.03$) than did root-free soil ($0.78 \pm 0.02$; one-way ANOVA, $p=0.029$). Furthermore, ARQ of the soil incubations significantly declined with depth from $0.82 \pm 0.01$ at 0-20 cm to $0.74 \pm 0.02$ at 20-40 cm (one-way ANOVA, $p=0.0053$).



## 4 Discussion

There are many factors that can affect ARQ; however, our evidence indicates the strong seasonal patterns in ARQ and $\delta^{13}CO_2$ were likely driven by changes in the amount of root-derived organic substrates providing energy for heterotrophic microbial respiration and changes in the contributions of autotrophic root respiration. This interpretation is supported by previous soil ARQ studies, our incubations, and the scientific understanding of how plant carbon inputs change seasonally. The seasonal range in ARQ from ≈0.9 during the growing season to ≈0.7 during the winter may reflect a shift in the molecules fueling respiration from more oxidized substrates like sugars and organic acids derived from roots in the summer to more reduced substrates in the winter such as lipids and proteins derived from microbial necromass. Previous incubations found that glucose additions increased ARQ (Dilly, 2001; Theenhaus et al., 1997). Other studies attributed a decline in ARQ during the time course of incubation to the depletion of labile carbon sources (Angert et al., 2015; Severinghaus, 1995). Our short-term incubations demonstrated that root respiration has a greater ARQ than microbial respiration from root-free soils. During the growing season, root respiration and exudation increase, which should increase ARQ, as seen in our data. In Eastern U.S. deciduous forests, root exudation rates tend to be lower in the winter and spring than in the summer and fall (Abramoff and Finzi, 2016; Phillips et al., 2008). Mass-specific fine root respiration rates were greater during the growing season (up to 8 nmol $CO_2$ $g^{-1}$ $s^{-1}$) than in the winter (<1 nmol $CO_2$ $g^{-1}$ $s^{-1}$) and total belowground carbon flux was greatest from May through October (Abramoff and Finzi, 2016). Though these root studies were not from the western United States, eddy covariance data from a coniferous forest near our study site found that primary production was greatest during the summer months from June through mid-September (Goldstein et al., 2000).

Beyond the results of our root and root-free soil incubations, there is additional evidence that root and rhizosphere respiration should have a greater ARQ than microbial-derived respiration. For example, respiration of root tips is driven by sugar content and has an RQ of 1.0 (Saglio and Pradet, 1980). Furthermore, recent metabolomic analysis of root exudates identified sugars, carboxylic acids, amino acids, and phenolics as the main metabolites (Zhalnina et al., 2018), most of which are relatively oxidized energy sources with greater respiratory quotients. Thus, we would expect greater ARQ values during the summer due to higher root activity. When trees are dormant, the lack of fresh inputs from roots may lead to more recycling of organic carbon within microbial biomass, wherein proteins and lipids are the first and third largest constituents by weight, making up to 55% and from 10-35% of a typical bacterial cell's dry mass, respectively (Kleber and Reardon, 2017; Neidhardt, 1987). Lipids and proteins tend to be reduced and have the lowest RQ values of common organic substrates, likely explaining the lower wintertime ARQ values in our soils.

The seasonal pattern in $\delta^{13}CO_2$ reinforces our interpretation that changes in respiration carbon sources were driving changes in ARQ. Soil $\delta^{13}CO_2$ was more enriched in the summer and became more depleted in the winter by up to 2‰. In a comprehensive review of carbon isotopes in terrestrial ecosystems, Bowling et al. (2008) showed that plant lipids tend to be



more depleted in $^{13}C$ while sugars and organic acids tend to be more enriched in $^{13}C$ relative to bulk leaf $\delta^{13}C$. While these
numbers are based on plant lipids, if we assume microbial lipids are similarly depleted relative to other organic compounds,
an increase in microbial necromass as an organic matter source relative to root-derived sources during the winter would cause
the observed fluctuation in $\delta^{13}CO_2$. Furthermore, a chemical fractionation of soil organic matter found that the water-soluble
fraction, which includes sugars, was 3-4‰ more enriched than the acid-insoluble pool (Biasi et al., 2005). While the
interpretation of respiration $\delta^{13}C$ by itself in $C_3$ ecosystems can be difficult due to the small ‰ differences among carbon
sources (e.g., Bowling et al., 2015), the simultaneous use of ARQ and $^{13}CO_2$ helps strengthen interpretations.

Seasonality encompasses changes to phenology and soil climate. Both ARQ and $\delta^{13}C$ had significant positive relationships
with soil temperature. In addition to the importance of plant phenology described above, temperature could have direct effects
on respiration sources. Specifically, warmer temperatures can increase root exudation rates (Yin et al., 2013) and the relative
contribution of autotrophic-derived, if not directly autotrophic, respiration to total soil respiration. In two subarctic ecosystems,
warming increased the proportion of ecosystem respiration derived from autotrophs (which, using natural abundance
radiocarbon as a tracer, included heterotrophic respiration of root exudates) relative to heterotrophs (Hicks Pries et al., 2015).
However, temperatures can affect ARQ through more than just changing the contributions of autotrophic sources. Lower
temperatures increase the thermodynamic favorability of the oxidation of reduced carbon in compounds like lipids (LaRowe
and Van Cappellen, 2011), which could also explain the decrease in ARQ values at lower temperatures. For $\delta^{13}C$, it is likely
that phenological changes to organic carbon sources were more important than temperature per se. Several soil incubation
studies show that increases in temperature cause respired $\delta^{13}CO_2$ to become depleted by about 0.12–0.35‰ for each 1°C rise
in temperature—the opposite of the relationship we found (Andrews et al., 2000; Biasi et al., 2005; Hicks Pries et al., 2013).
In these incubations, which were devoid of new organic carbon inputs, unlike in situ conditions, the shift was attributed to
changes to the microbial community that affected carbon source preferences (Andrews et al., 2000; Biasi et al., 2005).
Furthermore, in a Mediterranean climate, phloem sap from trees has been shown to become more enriched in $\delta^{13}C$ during the
summer (Merchant et al., 2010), matching our pattern in soil $\delta^{13}CO_2$.

Soil temperature and soil moisture were so strongly negatively correlated due to our study site's Mediterranean climate that it
is difficult to separate their effects. ARQ and $\delta^{13}CO_2$ were negatively correlated with volumetric water content, which was
greatest when soil temperatures were coldest. Volumetric water content has the potential to control ARQ in several ways. First,
increased soil moisture reduces $O_2$ availability, which could increase ARQ values >1 as $CO_2$ is produced without additional
$O_2$ consumption. However, during our study the soil remained oxic (soil $O_2$ averaged 20% and the minimum was 17.38%).
The negative relationship between ARQ and soil moisture indicates that anaerobic respiration was not a driver, and we only
measured one ARQ value greater than one (1.03) during our study. However, diffusion rates are lower with higher soil
moisture, which could make detection of high ARQ values difficult if anoxic conditions occur within microaggregates. In
anoxic microaggregates, iron (II) is produced anaerobically, which is subsequently oxidized to iron (III) as the aggregate dries



and becomes aerobic, a process that consumes $O_2$ without producing $CO_2$, resulting in low ARQ values that can be detected
as drying soils increase diffusion (Angert et al., 2015). In our soils, which tend to contain relatively high amounts of iron
oxides (Rasmussen et al., 2005), iron oxidation could explain the 15% of ARQ values that were less than the reduced organic
matter value of 0.7. Lastly, since $CO_2$ is more soluble in water than is $O_2$, more $CO_2$ relative to $O_2$ is expected to dissolve in
soil water, which would reduce ARQ values at higher moisture contents. However, different dissolution rates and iron
oxidation do not fully explain our data as the wide variability in ARQ values (0.44 to 0.94) at high volumetric water contents
(0.27 to 0.31) can be best explained by time of year (Fig. A1), which again points to phenology as the main driver; the greater
ARQ values are from April and June while the lower values are from February and March. Furthermore, there was a stronger
relationship between observed and predicted ARQ in the temperature model than from the soil moisture model.

The reasons for $\delta^{13}CO_2$ becoming more depleted with increasing volumetric water content are not clear. Based on kinetics, we
would expect that as more $CO_2$ dissolves in water, the soil air should become enriched in $^{13}CO_2$ because dissolution
discriminates against the heavy isotope and increasingly so at lower temperatures (Zhang et al., 1995), but our data were not
consistent with this explanation.

The warming treatment caused slightly greater ARQ values (only in the winter) and slightly more enriched $\delta^{13}CO_2$ relative to
the controls; trends that are similar to those followed by ARQ and $\delta^{13}CO_2$ with soil temperature. The warming treatment effect
in terms of ARQ and $\delta^{13}CO_2$ was relatively small (0.07 units and 0.48‰), but its effect on soil carbon flux was large. Warming
increased $CO_2$ production by 34 to 37% with about 50% of the respiration and 40% of the warming response occurring below
15 cm in the soil profile (Hicks Pries et al., 2017).

The direction of the shift in ARQ and $\delta^{13}CO_2$ with warming to slightly higher ARQ and more enriched $\delta^{13}CO_2$ values indicates
proportionately more respiration of relatively oxidized, labile organic substrates. In principle, the shifts in ARQ and $\delta^{13}CO_2$
could also result from a relative increase in availability of labile organic substrates, perhaps as a result of enhanced root growth
and exudation (Yin et al., 2013). Given that ARQ only differed by treatment in the winter when trees were less active, however,
the former situation of preferential decomposition seems more likely. If so, this could lead to exhaustion of that pool and
eventually lower warming-induced $CO_2$ losses as seen at Harvard Forest (Melillo et al., 2002, 2017). Further measurements of
$CO_2$ production, ARQ, and $\delta^{13}CO_2$ as warming progresses will help distinguish these cases.

Depth was the only parameter by which ARQ and $\delta^{13}CO_2$ did not change in concert with one another. ARQ decreased with
depth while $\delta^{13}CO_2$ became more enriched. The decrease in ARQ with depth, which was more dramatic in the root-free soil
incubations than in soil air (difference of 0.08 versus 0.03), is likely due to decreased plant inputs with fewer fine roots and
less root exudation at depth (Hicks Pries et al., 2018; Tückmantel et al., 2017). The enrichment of soil $\delta^{13}CO_2$ likely reflects
the near-universal enrichment of soil organic carbon with depth due to catabolic carboxylation reactions (as microbial





byproducts and necromass become a larger proportion of soil organic matter; Ehleringer et al., 2000; Torn et al., 2002) or the
Suess effect (the continuing depletion of atmospheric $CO_2$ over time due to the burning of fossil fuels). In our soils, there was
about a 2‰ enrichment in bulk soil organic $\delta^{13}C$ with depth (Hicks Pries et al., 2018).
**5 Conclusion**
Here we have shown, for the first time, both annual patterns in soil ARQ and how ARQ is affected by experimental warming.
The seasonal patterns in ARQ were likely due to changes in the substrates providing the energy for soil respiration with root-
derived sugars and organic acids being the dominant substrates during the growing season and microbial necromass being the
dominant substrate during the winter. Our inferences of organic substrates based on ARQ were supported by soil $\delta^{13}CO_2$
measurements, which showed clear patterns despite our study system containing only $C_3$ plants. We have shown how ARQ
measurements can help to disentangle the biological sources contributing to soil respiration and to understand how sources are
shifting due to global change. This application of ARQ worked well in our soils, which were well-drained, oxygenated, and
lacked carbonates. The interpretation of soil ARQ values becomes more complex if those conditions are not met (Angert et
al., 2015). The autotrophic and heterotrophic source separation in our incubations indicates ARQ has the potential to be used
to partition soil respiration in a similar manner to natural abundance $\delta^{13}C$ (e.g., Dorrepaal et al., 2009; Hicks Pries et al., 2013).
To enable further applications of ARQ, more characterization is needed of the controls of the ratio, including incubation studies
of sterile and 'live' soils under aerobic and anaerobic conditions and co-located measurements of ARQ fluxes and the oxidative
ratio of organic matter sources as in Masiello et al. (2008). Such future investigations will help determine whether ARQ
deserves a prominent place alongside natural abundance isotopes in the ecosystem ecology and biogeochemistry toolkit.
**Data Availability**
Data is being made publicly available on ESS-DIVE (http://ess-dive.lbl.gov/).
**Author Contribution**
CHP, AA, and MST conceived of the study. Field measurements were conducted by CHP and CC. Lab analyses were
conducted by CHP and BH. Statistical analyses were conducted by CHP. CHP wrote the manuscript with feedback from all
authors.
**Acknowledgements**
This work was supported as part of the Terrestrial Ecosystem Science Program by the Director, Office of Science, Office of
Biological and Environmental Research, of the U.S. Department of Energy under Contract No. DE-AC02-05CH11231. We



would like to acknowledge Rachel Porras for her assistance running the isotopic samples, and Bryan Curtis and Biao Zhu for
their contributions to setting up the warming experiment.

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






| Molecule | RQ[a] | $\delta^{13}C$ (relative to bulk leaf)[b] |
|---|---|---|
| Organic acids | 1.4 (1-4) | +0.75 |
| Sugars | 1.0 | +1.5-2 |
| Phenolics | 0.95 | NA |
| Proteins | 0.77 (0.67-1.0) | +1 |
| Lignin | 0.88 | -3 |
| Lipids | 0.73 | -4 |

[a] Data from Masiello et al. 2008
[b] Data from Bowling et al. 2008
**Table 1. Respiration quotient (RQ) and relative isotopic enrichment of common molecules/substrates for respiration found in soils.**





| Depth (cm) | $\delta^{13}CO_2$ (‰) | ARQ |
|---|---|---|
| 30 | -29.0 ± 0.09 | 0.84 ± 0.02 |
| 50 | -28.6 ± 0.08 | |
| 70 | -28.4 ± 0.07 | |
| 90 | -28.3 ± 0.08 | 0.81 ± 0.02 |

**Table 2. Mean (± SE) corrected $\delta^{13}CO_2$ and ARQ of soil pore space by depth.**


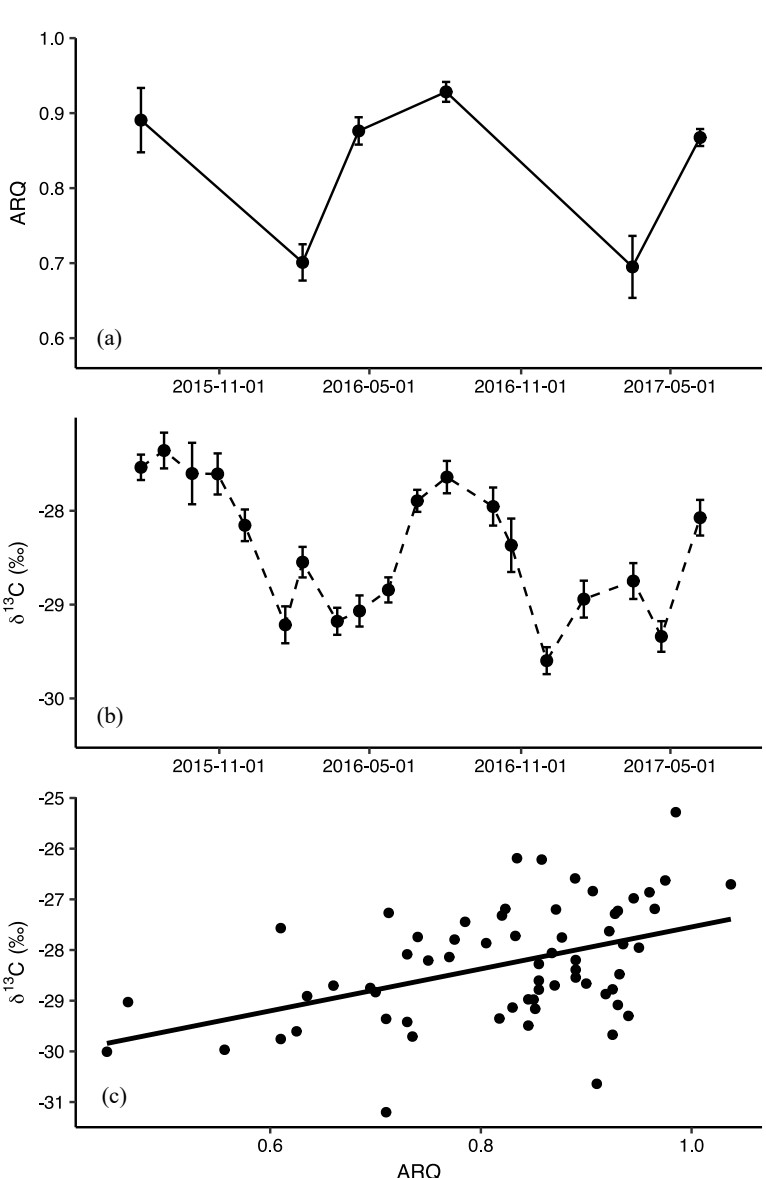


**Figure 1. Mean (± SE) apparent respiration quotient (ARQ; a) and corrected $\delta^{13}CO_2$ (b) in soil pore space air averaged across all depths and treatments by sampling month. The relationship between ARQ and $\delta^{13}CO_2$ values over the months when they were sampled simultaneously (c). The line shows the fit of a mixed model regression where gas well was treated as a random effect.**







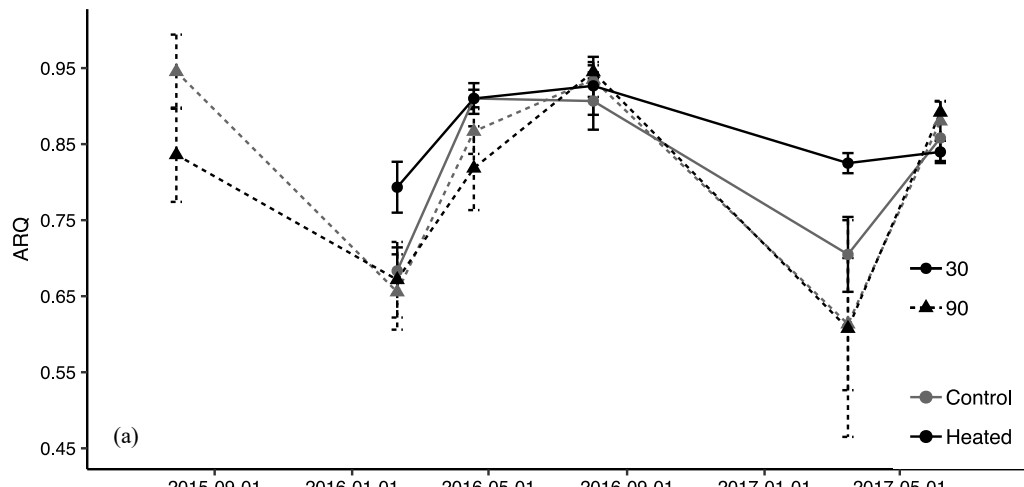


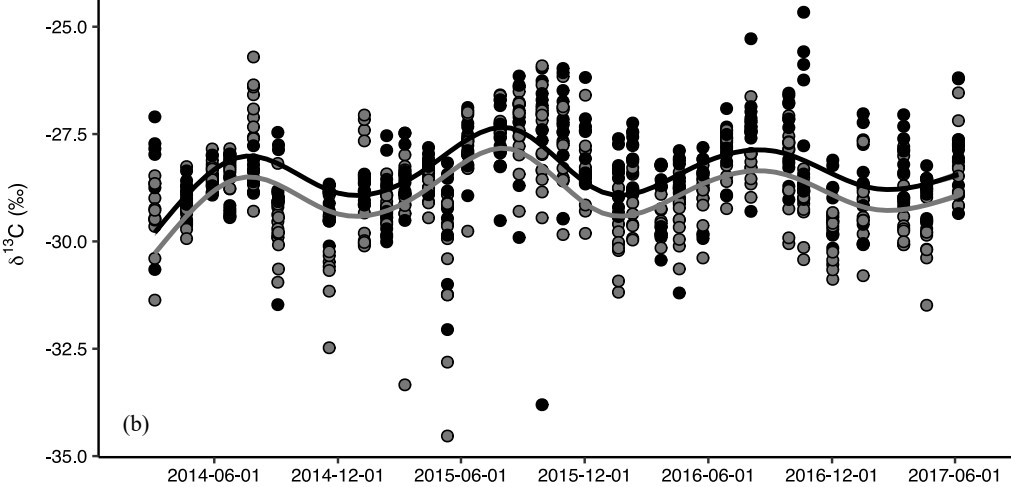



**Figure 2. Mean (± SE) apparent respiration quotient (ARQ) by sampling date for heated (black) and control (grey) treatments at 30**
**cm (circles) and 90 cm (triangles) depths (a). ARQ only differed significantly among treatments during the winter at 30 cm.**
**Corrected $\delta^{13}CO_2$ for all depths (30, 50, 70, and 90 cm) and months sampled (b). The lines represent the predicted fit of a sinusoidal**
**regression (see text) for an average soil depth in control (grey) and heated (black) treatments.**


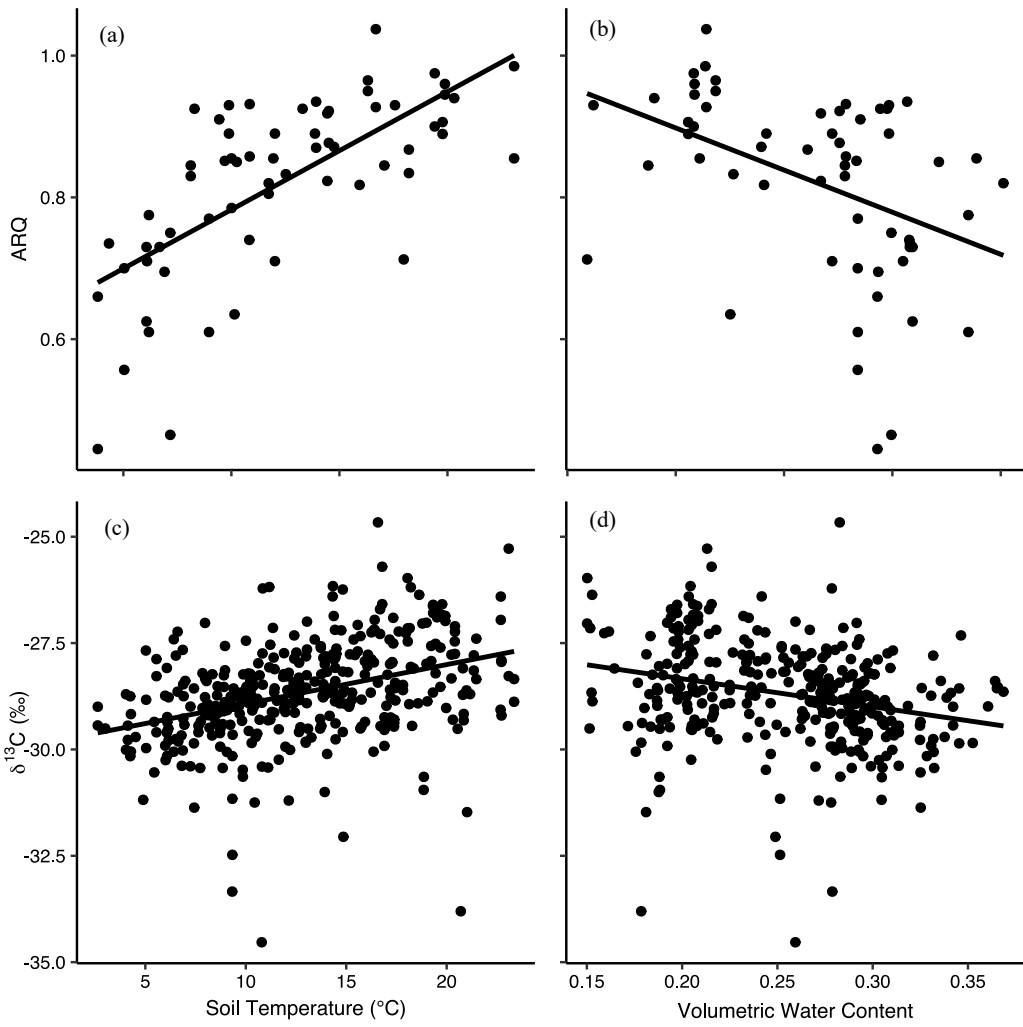


**Figure 3. The relationships of apparent respiration quotient (ARQ; a, b) and $\delta^{13}CO_2$ (c, d) by soil temperature (a, c) and soil moisture (b, d). The points represent data taken from an individual gas well. The lines show the fit of a mixed model regression between each variable where individual gas well was treated as a random effect.**








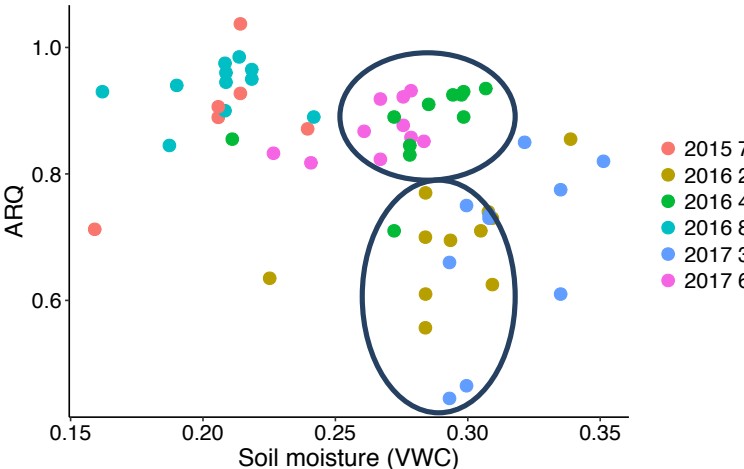


**Figure A1. The variability in the apparent respiration quotient's (ARQ) relationship with soil moisture at 0.25 to 0.30 VWC can be explained by time of year. The greater ARQ values are from April and June while the lower values are from February and March. Colors are by sampling date shown as year and month.**

