# Peer review of "Using Respiration Quotients to Track Changing Sources of Soil Respiration Seasonally and with Experimental Warming"

_Biogeosciences, 2019_

## Referee Comment (RC1) · Anonymous Referee #1 · 5 Jul 2019

General comments

This manuscript describes an analysis of field-measured apparent respiration quotient (ARQ), along with d13CO2 data, from a soil warming experiment. The promise of such data, as is well explained in the introduction, is the insight it offers into changing sources of soil respiration at a much finer resolution than simply heterotrophic vs. autotrophic. Whether ARQ will live up to this promise is uncertain, but this ms is a significant step in that direction. The authors' results, which documented changes in ARQ with temperature, moisture, and most dominantly season, are intriguing and mostly consistent with theoretical expectations. One could imagine extensions to the experiment–most obviously to me, it's too bad no measurements were made on root-free (trenched) plots–but this is a really nice study, well written, and compelling.

There are some problems. The introduction needs to be careful about ARQ versus RQ usage; I have some concerns about the statistics, which seem a bit ad hoc at times; there's currently no code or data availability, which I view as unacceptable; and some of the figures and tables need minor clarifications. See comments below for more detail.

Overall, however, this is a strong and interesting study and will pique the interest of a large group of soil researchers interested in discriminating soil respiration sources.

Specific comments

1. Line 33: kind of true but not exactly; typically defined as the surface-to-atmosphere $CO_2$ flux (yes, dominated in most systems by RH+RA). Worth clarifying I'd suggest

2. L. 43: might cite e.g. Subke et al. (2006, http://dx.doi.org/10.1111/j.1365-2486.2006.01117.x) or Bond-Lamberty et al. (2004, http://dx.doi.org/10.1111/j.1365-2486.2004.00816.x) here?

3. L. 48: 1/RQ or 1/ARQ? Is there a difference between ARQ and RQ? Also not sure why OR is defined here as it doesn't seem to be used again

4. L. 49-50: is this (carbohydrate=1) defined as a standard, or does it follow from elemental structure?

5. L. 55: ah. So here define ARQ; line 48 should solely reference RQ, then

6. L. 82: "same soils"

7. L. 84: probably start new paragraph

8. L. 167-169: I know this is standard but it's also unclear and hard to replicate; in the future consider using something like the MASS::stepAIC() function, which automates this term-selection process in a transparent and reproducible way

9. L. 170: start new paragraph

10. L. 180: hmm, consider something like piecewiseSEM::rsquared() which will do this more robustly and statistically appropriately

11. Data AND code availability? I generally expect both to be available (attached as SI or deposited) for transparency and reproducibility, at least for the main results . . .line 339 is nice but not sufficient

12. Table 2: define ARQ in caption and explain difference from RQ in table 1

13. Figure A1: what are the circles?

14. L. 238-: nicely written

15. L. 260: "climate, among other factors."

16. L. 296-: hmm

17. L. 312: SOC losses?

18. L. 331-332: this probably deserves a bit more discussion, in the discussion or introduction

19. L. 336-337: good close!

---

## Author Comment (AC1) · 12 Jul 2019

Thank you for the helpful feedback. I am currently working on uploading the data to a repository and will clean up my R code to include it as a supplement.

---

## Referee Comment (RC2) · Anonymous Referee #2 · 4 Aug 2019

This paper reports on a soil warming experiment in a forest in California and its effects on soil respiration processes using stable isotopes and an index of the ratio of CO2 to O2 in soil air (ARQ) that accounts for differences in diffusivity. The paper explores patterns in these values, rather than presenting a hypothesis testing framework. The results show a nice time series with interesting seasonal patterns that are interpreted to indicate changing substrate source for respiration. Asserting that different organic compounds with different oxidative states are the main explanation for the observed patterns is speculative. The discussion and conclusion should acknowledge the sources of uncertainty more explicitly, specifically that the RQ can be quite variable for different organic compounds. L 151-152, where were ambient CO2 samples collected

(what height above soil surface)? Please provide a reference for O2 concentration of the atmosphere. L 190, L 250, L316, L318, and throughout, please refer to delta values as higher or lower rather than enriched or depleted, because the latter terms refer to the relative differences of 13C in molecules whereas delta values are ratios. The rest of the discussion paragraph on L 251-258 uses enriched/depleted terminology correctly. L. 197-205. This paragraph would be easier to understand if you could use more consistent sentence structure for each sentence. Please provide the ANOVA table so readers can just see the results all in one place. L. 197 has a mistake in the table number. L 207-213. Did you look for an effect of treatment on the relationships between soil climate and ARQ or ïẠd'13C? Since there was a treatment-by-date interaction for ARQ this might indicate that warming is altering the temperature sensitivity (hint of this in Fig 2a for 30 cm). I understand that you didn't report on treatment-by-date interaction for ïẠd'13C due to the significant autocorrelation but there might also be some effect of treatment on the temperature and moisture relationships shown in Fig 3. Why not just run regressions for the warming and control treatments separately? This seems to be a missed opportunity for a potentially stronger paper. L. 215-218. What is the sample size for the ARQ measurements in roots and root-free soil? What were the ïẠd'13C values in these samples? L 242, greater RQ's than what? L 296-299, increasing ïẠd'13C values with decreasing soil water content might be associated with larger advection of atmospheric CO2 into the soil which is not accounted for by the diffusion correction. This is probably also the reason for having to throw out the data from 15-cm depth (effective diffusion fractionation is <4.4 permille due to advection effects, so applying the diffusion correction equation leads to values that are unrealistically low).

Data availability: please provide a more specific url for the data or provide it in a supplement.

Table 1. If possible, provide estimated ranges of these values for all compounds for both ARQ and ïẠd'13C, using a wider lit review if needed. This variability is a major source of uncertainty in the interpretations of the ARQ results but is not mentioned in

the text.

Table 2. Provide sample size (n) for the values, including more information in the caption (presumably these were collected over the duration of the study).

Fig. 1. Give sample sizes for ARQ and ïĄd'13C in caption (I think it's n=6 plots * 4 depths for ïĄd'13C, and n = 6 plots * 2 depths for ARQ per date?). Great to have so many samples! But it's important to readers to appreciate the number of samples that went into the SE calculation. Connecting the points in the ARQ plot is not very meaningful considering the sparse sampling dates. Provide fitting statistics for the regression in part c; even though these are given in the text, it's nice to have this info in the figure so readers don't need to scour the text to find out the important details.

Fig 2. This figure is a bit odd because it plots time series data for the two constituents using means for ARQ and raw data for ïĄd'13C. It's nice to see the datapoints in part b of this figure. Please include actual data in part a also, and plot them on the same time scale.

---

## Author Comment (AC3) · 31 Jan 2020

We agree that our attribution of changes to ARQ to different substrates is speculative. We added this sentence to the manuscript to highlight the uncertainty: "We caution that direct experimental evidence of how ARQ changes with sources is needed before our inferences of substrate use can be proven." we also now report ranges of ARQ values for each substrate type in Table 1.

We added the height of ambient air sampling (0.5 to 1 m aboveground).

We added a citation for the atmospheric concentration of oxygen.

We changed enriched and depleted to higher/greater and lower when referring to delta values. We rewrote that results paragraph with better parallel construction of the sentences. Since these were mixed model multiple regressions, we do not have traditional ANOVA tables to report.

We fixed the table number.

The suggestion that we investigate the effect of the warming on temperature sensitivity was a great one. We re-ran our linear models of ARQ and 13CO2 vs climate (soil T or VWC) with a climate by warming treatment interaction, which improved the model in all cases except for one (13CO2 vs VWC). The reviewer was correct in stating that this was a missed opportunity. These new analyses make a much stronger case for the experimental warming treatment affecting the substrates being used to fuel soil respiration. We have thus expanded our discussion section to include more about how warming and soil moisture can affect root respiration and substrate utilization.

We added the sample size (n=4) for the incubations. We did not have 13C values as we had issues with our Picarro at that time.

We reworded line 242.

We added your comment about advection possibly leading to erroneous diffusive fractionation corrections to the manuscript.

The data (doi:10.15485/1596312) are available on ESS DIVE now.

We added a range of RQ values for the molecules in table 1.

We have added sample sized to all the captions of all tables and figures.

In figure 1, we chose to keep the lines connecting the ARQ points as the point of figure 1 is to show how the ARQ and 13C patterns are similar. We added the fit statistics to the caption.

In figure 2, we changed the x-axis so they are the same for both panels, and, as

requested, we include all the data points for the ARQ data as well.

---

## Author Response (AR1)

**Reviewer 1 Response**

We are pleased to report that we have included our R script for the analyses in this paper as part of the supplemental information. We have also made sharing R script sharing a policy for papers produced in my lab going forward. The data in this manuscript are now available on ESS DIVE (doi:10.15485/1596312).

We edited our second paragraph slightly. We still wanted to introduce apparent after respiration quotients are defined in their "ideal" form so as not to confuse the reader.

Specific changes:

1: Soil respiration was redefined.
2: Cited Subke et al. 2006
3: Removed definition of OR and cleared up ARQ/RQ confusion.
4: Based on its elemental structure
5: Fixed
6: Added "same"
7: Started new paragraph.
8: I am not a fan of MASS: stepAIC() as I find it is too conservative in regards to the predictor variables it leaves in a model. I prefer to remove variables when their removal does not change the AIC or significantly change the results of the LRT based on Occam's razor. Anyways, the R code is now available so it can be replicated.
9: Started new paragraph.
10: We recalculated using this command, R squared were either the same or within 0.02. Great tip, thank you!
11: The R code is now SI. The data have been submitted to ESS DIVE.
12: Changed.
13: Defined the circles in the caption.
14: Thanks
15: Added.
16: ?
17: Yes, SOC. Good catch.
18: Since those conditions were met in our soils, we do not think discussing it further is relevant to this manuscript. There is a good discussion of it in the Angert paper we cited.
19. Thanks

**Reviewer 2 Response**

We added the height of ambient air sampling (0.5 to 1 m aboveground).

We added a citation for the atmospheric concentration of oxygen.

We changed enriched and depleted to higher/greater and lower when referring to delta values.

We rewrote that results paragraph with better parallel construction of the sentences. Since these were mixed model multiple regressions, we do not have traditional ANOVA tables to report.

We fixed the table number.

The suggestion that we investigate the effect of the warming on temperature sensitivity was a great one. We re-ran our linear models of ARQ and 13CO2 vs climate (soil T or VWC) with a climate by warming treatment interaction, which improved the model in all cases except for one (13CO2 vs VWC). The reviewer was correct in stating that this was a missed opportunity. These new analyses make a much stronger case for the experimental warming treatment affecting the substrates being used to fuel soil respiration. We have thus expanded our discussion section to include more about how warming and soil moisture can affect root respiration and substrate utilization.

We added the sample size (n=4) for the incubations. We did not have 13C values as we had issues with our Picarro at that time.

We reworded line 242.

We added your comment about advection possibly leading to erroneous diffusive fractionation corrections to the manuscript.

The data (doi:10.15485/1596312) are available on ESS DIVE now.

We added a range of RQ values for the molecules in table 1.

We have added sample sized to all the captions of all tables and figures.

In figure 1, we chose to keep the lines connecting the ARQ points as the point of figure 1 is to show how the ARQ and 13C patterns are similar. We added the fit statistics to the caption.

In figure 2, we changed the x-axis so they are the same for both panels, and, as requested, we include all the data points for the ARQ data as well.

**Using Respiration Quotients to Track Changing Sources of Soil Respiration Seasonally and with Experimental Warming**

Caitlin Hicks Pries[1,2], Alon Angert[3], Cristina Castanha[2], Boaz Hilman[3,4], and Margaret S. Torn[2]

[1]Department of Biological Sciences, Dartmouth College, Hanover, NH, 03784, United States of America

[2] Climate and Ecosystem Science Division, Earth and Environmental Science Area, Lawrence Berkeley National Laboratory, Berkeley, CA, 94720, United States of America

[3]The Institute of Earth Sciences, The Hebrew University of Jerusalem, Givat-Ram, Jerusalem 91904, Israel

[4]Currently at: Department of Biogeochemical Processes, Max-Planck Institute for Biogeochemistry, Jena, 07745, Germany

*Correspondence to*: Caitlin Hicks Pries (Caitlin.Pries@dartmouth.edu)

**Abstract.** Developing a more mechanistic understanding of soil respiration is hampered by the difficulty in determining the contribution of different organic substrates to respiration and in disentangling autotrophic versus heterotrophic and aerobic versus anaerobic processes. Here, we use a relatively novel tool for better understanding soil respiration: the apparent respiration quotient (ARQ). ARQ is the amount of $CO_2$ produced in the soil divided by the amount of $O_2$ consumed and it changes according to which organic substrates are being consumed and whether oxygen is being used as an electron acceptor. We investigated how the ARQ of soil gas varied seasonally, by soil depth, and by *in situ* experimental warming (+4°C) in a coniferous forest whole-soil-profile warming experiment over two years. We then compared the patterns in ARQ to those of soil $\delta^{13}CO_2$. Our measurements showed strong seasonal variations in ARQ from ≈0.9 during the late spring and summer to ≈0.7 during the winter. This pattern likely reflected a shift from respiration being fueled by oxidized substrates like sugars and organic acids derived from root and root respiration during the growing season to more reduced substrates such as lipids and proteins derived from microbial necromass during the winter. This interpretation was supported by $\delta^{13}CO_2$ values, which were relatively lower, like lipids, in the winter and relatively higher, like sugars, in the summer. Furthermore, experimental warming significantly changed how both ARQ and $\delta^{13}CO_2$ responded to soil temperature. Wintertime ARQ and $\delta^{13}CO_2$ values were higher in heated than in control plots, probably due to the warming-driven increase in microbial activity that may have utilized oxidized carbon substrates, while growing season values were lower in heated plots. Experimental warming and phenology change the sources of soil respiration throughout the soil profile. The sensitivity of ARQ to these changes demonstrates its potential as a tool for disentangling the biological sources contributing to soil respiration.

Field Code Changed
Field Code Changed
Field Code Changed
Field Code Changed
Field Code Changed
Field Code Changed
Field Code Changed
Field Code Changed
Field Code Changed
Field Code Changed
Field Code Changed
Field Code Changed
Field Code Changed
Field Code Changed
Field Code Changed

[revised manuscript text omitted]